# Study on Hypervelocity Impact Characteristics of Ti/Al/Mg Density-Graded Materials

**Luping Long [1], Yingbiao Peng [1,\*], Wei Zhou [1] and Wensheng Liu [2]**

1    College of Metallurgical and Materials Engineering, Hunan University of Technology, Zhuzhou 412007, China; zmr_llp@163.com (L.L.); zhouwei@hut.edu.cn (W.Z.)
2    State Key Laboratory of Powder Metallurgy, Central South University, Changsha 410083, China; liuwensheng@csu.edu.cn
\*    Correspondence: pengyingbiao1987@163.com; Fax:+86-0731-2218-3461

**Abstract:** An improved shielding structure of a bumper that constructed from Ti/Al/Mg density-graded materials was presented. Two types of Ti/Al/Mg density-graded materials with the same areal density were prepared by diffusion bonding and powder metallurgy, respectively. The characteristics of hypervelocity impact including penetration holes in the bumper, damage patterns on the rear wall and micrographs of the crater were investigated. The results show that damage mechanism of Ti/Al/Mg density-graded materials is closely related to the interface bonding strength and matrix strength. The penetration holes of Ti/Al/Mg density-graded material obtained by diffusion bonding exhibit typical ductile characteristics. The Ti/Al/Mg density-graded material prepared by powder metallurgy shows significant mechanical synergistic response under high strain compression and appears fragile characteristic. The shielding performance of Ti/Al/Mg bumper is increased by 20.4% compared with aluminum bumper. A theoretical analysis suggests that a Ti-Al-Mg bumper can fully break the projectile and greatly increase the entropy during the impact process. Larger projectile kinetic energy is converted into the internal energy during the impact process, thereby causing an obvious increase in shielding performance.

**Keywords:** Ti/Al/Mg density-graded material; diffusion bonding; powder metallurgy; hypervelocity impact characteristics

## 1. Introduction

With the rapid development of aerospace technology, the debris environment has been worsening, and the service life of orbiting spacecraft has been greatly restricted [1]. For space debris of a millimeter level and below, protective structures composed of high-tech materials can be set in several key parts [2]. At present, there have been a limited number of high-performance protective materials, such as Kevlar cloth, Nextel cloth, Beta cloth, so it is necessary to develop a new protective material [3–9]. Over the recent decades, the homogeneous single-layer sheet has been replaced by multi-layer density-graded material sheet, such as Ti/Al/nylon [10] and Al/Mg [11]. The density gradient variation in the thickness direction to achieve high levels of impedance mismatch has been reported to improve penetration resistance through by multiple interface reflections and transmissions [12,13]. In previous studies on the hypervelocity impact characteristics of materials, most of the studies concern traditional titanium, aluminum, iron and flexible materials, mainly based on the mechanical point concerning impact vaporization, jetting process, stress-wave propagation and the thermodynamic state of impact fragment [14,15]. In this study, Ti/Al/Mg density-graded materials were prepared by diffusion bonding and powder metallurgy, respectively. Deformation and damage characteristics of these materials

under hypervelocity impact were systematic analyzed. The impact damage mechanism of Ti/Al/Mg density-graded materials can be revealed by damage morphologies.

## 2. Material and Experimental Procedure

### 2.1. Material Preparation

The Ti/Al/Mg density-graded material prepared via diffusion bonding in this study was recorded as sample I. Ti-TC4 bar, Al-2A12 plate and Mg-AZ31 bar were used as raw materials. The Ti/Al joint was achieved at 650 °C for 2 h under the pressure of 10 MPa, while Al/Mg was bonded at 475 °C for 1.5 h under the pressure of 6 MPa. Sample II was prepared by powder metallurgy technology, which includes multi-step powder metallurgy pre-sintering and interfacial joining. The particle size of Ti-6Al-4V mixed powder, Al-2A12 and Mg-AZ31 alloy powder were 45, 58 and 48 μm, respectively. Ti-6Al-4V was pre-sintered under a hot-pressing condition under 5 MPa at 1250 °C for 3 h, Al-2A12 was pre-sintered at 550 °C for 2 h, while Mg-AZ31 was pre-sintered at 525 °C for 4 h. The Ti/Al joint was achieved at 625 °C for 3 h under the pressure of 4 MPa, while Al/Mg was bonded at 460 °C for 1.5 h under the pressure of 3 MPa. The thickness of samples I and II was 1.8 mm with a diameter of 80 mm. The areal density was equivalent to an aluminum alloy plate with the thickness of 1.5 mm.

### 2.2. Hypervelocity Impact Testing

Based on the Whipple structural protection Principle [16], aluminum alloy bumpers were replaced by sample I and sample II, respectively. A double-layer protective structure composed of bumper and Al-5A06 rear wall (simulated bulkhead) was formed, with the distance of 100 mm between them. In addition, an observation board consisting of Al-2A12 was performed at an interval of 25 mm. The impact tests at the velocity of 3.5 km/s were carried out on a two-stage light-gas gun, using Al-2A12 spheres with the diameter of 4.25 mm.

### 2.3. Analysis and Testing

Specimens with the dimensions of 10 mm × 10 mm × 1.8 mm were prepared from the Ti/Al/Mg density-graded materials by wire cutting, and multi-stage grinding by 200, 600, 1000, 1500 and 2000 grit SiC paper, then polished by 3 μm of $Al_2O_3$. The secondary electron images of interfaces were obtained by Nova Nano 230 scanning electron microscope (SEM, FEI, Hillsboro, OR, United States) equipped with the energy dispersive x-ray (EDS) detector. The tensile strengths of the matrixes were tested by the Instron3369 type mechanical testing machine (Instron, Canton, MD, United States). The tensile properties of the specimens with a 25 mm length were determined based on GBT228-2002, with a loading rate of 1 mm/min.

The macro damage morphology of bumper and the rear wall were analyzed by Canon EOS 5D4 digital camera (Canon, Tokyo, Japan) to record the macro damage to record the damage characteristic parameters of the bumper, such as the diameter of penetration holes and mass loss. Samples near the impact region of the projectile were cut and further analyzed by SEM with backscatter electron image.

## 3. Microstructure of Ti/Al/Mg Density-Graded Materials

Figure 1 shows the micrographs of Ti/Al/Mg density-graded materials. Both samples are well joined, without defects such as holes and discontinuities. The element concentrations across the Ti/Al and Al/Mg interfaces indicate that different intermetallic compounds are generated. The thickness of the Ti/Al interface layer in sample I is about 5 μm, while about 80 μm for the Al/Mg interface. Our previous studies have shown that the generated phases in the Al/Mg interface are $Al_3Mg_2$ and $Al_{12}Mg_{17}$ brittle intermetallic compounds [17]. There are holes in the Mg-matrix of sample II, which reduce the density and increase the density gradient of the entire material, however, decrease the matrix strength.

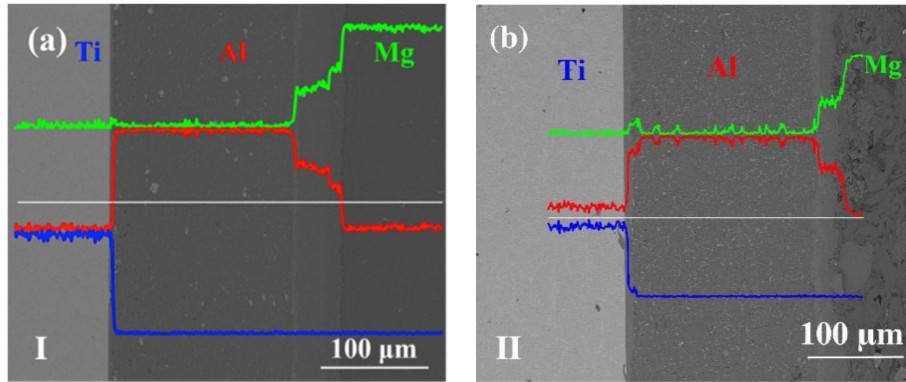

**Figure 1.** SEM micrographs at the interface of Ti/Al/Mg density-graded materials. (**a**) bumper I; (**b**) bumper II.

## 4. The Impact Characteristics of Ti/Al/Mg Density-Graded Materials

### 4.1. Macro Damage Characteristics

The macro-scale damage patterns of samples I and II are compared in Figure 2, where A and B denote the front surface and back surface, respectively. Sample I shows typical characteristics of a tough material, while sample II seems like a brittle material. The penetration hole diameters of two bumpers are 8.67 and 8.49 mm, and the mass losses are 1.30% and 1.24%, respectively. The Ti/Al composite layer in sample I shows a reverse petal-like failure pattern, and the Al/Mg interface is partially separated, apparently close to the crater. Sample II shows a regular circular penetration hole, and the crater of the Ti/Al layer has neat edges, with a few microcracks and tears. The back of the dumpers is also affected by shock waves, as shown in Figure 2b,d. Sample I forms bumps on the back, where cracks occurred. Due to the obvious brittleness of the Mg-matrix in sample II, it directly collapses via the shock wave. Moreover, the Al/Mg interface of sample I is completely delaminated by the lateral compression waves, while sample II is well bonded, just a slight delamination exists near the crater.

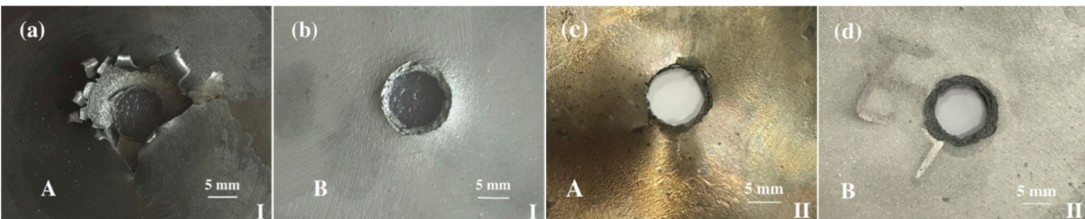

**Figure 2.** Macro-morphologies of the bumpers. (**a**,**b**) bumper I; (**c**,**d**) bumper II.

In general, compression shock waves determine macro damage patterns generated by the impact of the projectile on the Ti/Al/Mg bumper. When the incident shock wave contacts the front surface of the titanium alloy, the sparse waves are reflected from the side of the projectile and unloaded, and part of the titanium alloy splashes in the opposite direction via the tensile force generated by the sparse wave. A new sparse wave is reflected when the shock wave propagates into the Ti/Al interface. Shear force caused by the interaction of incident wave and sparse wave has been imposed to the interface, and the interface will be partially separated if the net shear stress exceeds dynamic fracture strength. Due to lateral sparse waves, the direction of interface separation is consistent with the crater outline. Therefore, it exhibits petal-like damage morphology. The shock wave propagates into the Al/Mg interface and continues to reflect the sparse wave, which leads to the delamination of the Al/Mg interface. Since the interface of sample II shows better performance, no delamination can be detected in sample II.

The damage of the rear wall is closely related to the assessment of protective performance. The damage characteristics of the rear wall with different dumpers are shown in Figure 3. It seems that the severely damaged regions are located at the center as a circular radial pattern. The rear wall is not penetrated in sample I, and there are many dense and overlap pits, spots and erosions on the front surface, seeing Figure 3a. The impact pits appear as blisters on the back surface without spalling. Small perforations occur which indicate failure, as shown in Figure 3c,d. The impact spots of sample II are denser than those of sample I. Perforations also produced on the rear wall, as shown in Figure 3e,f, indicating obvious failures. It can be proven that sample I and sample II have significantly better shielding performance than the aluminum alloy bumper. With the critical projectile diameter for 3.5 km/s, the shielding performance of sample I is increased by 20.4% in contrast with the aluminum bumper.

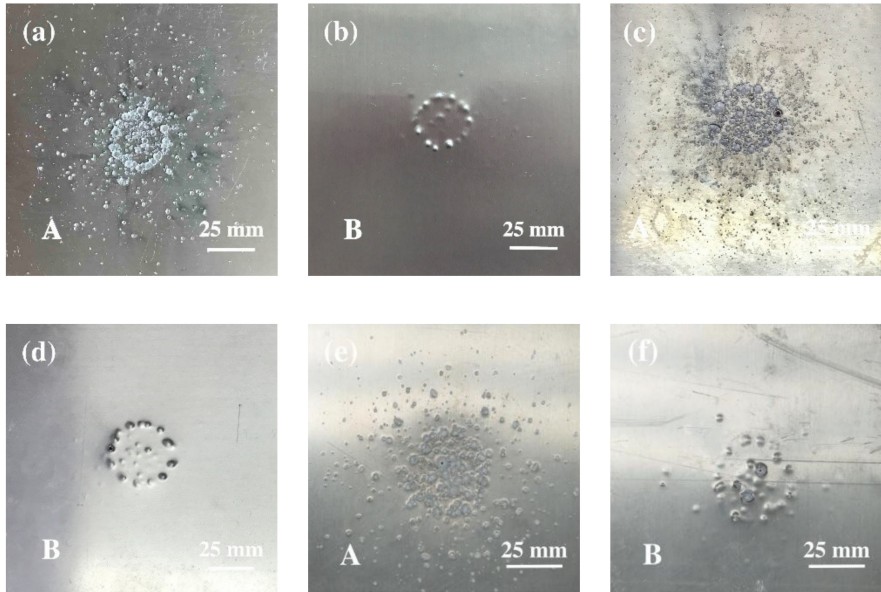

**Figure 3.** Macro-morphologies of the rear wall with different bumpers. (**a**,**b**) bumper I; (**c**,**d**) bumper II; (**e**,**f**) aluminum alloy bumper.

## 4.2. Micro Damage Characteristics

Due to the short acting time of the shock wave, the heat generated by impact near the crater wall cannot be effectively dissipated, so the most complicated microstructural changes occur at the crater wall. The micro-damage morphology of the crater wall in sample I is shown in Figure 4. It can be found that Ti, Al and Mg substrates all have brittle fracture characteristics under a high strain rate impact of the projectile.

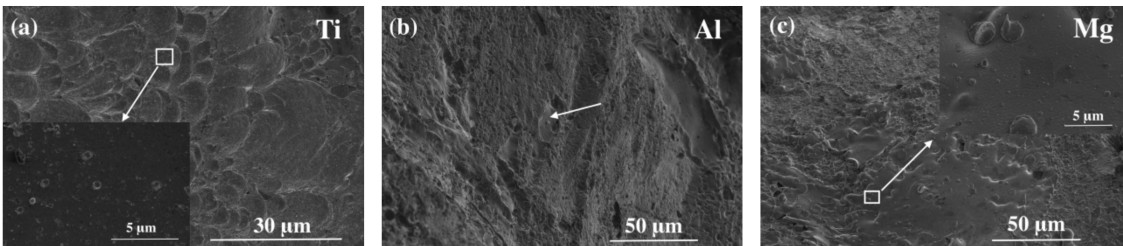

**Figure 4.** SEM micrographs of the crater for sample I. (**a**) Ti-matrix; (**b**) Al-matrix; (**c**) Mg-matrix.

Figure 4a shows many small craters with obvious directivity in the Ti-matrix, and the direction is consistent with the impact. From the enlarged area, it is clear that some small impact spots can be

detected in the crater. Therefore, it can be deduced that the main damage mechanism of the Ti-matrix is the erosion caused by the impact of projectile fragments. Some obvious erosion pits can also be detected in the Al-matrix, indicated by the arrows in Figure 4b. Moreover, microcracks at the edges of some large pits can be found. Shock crack propagates in the substrate, which causes the formation of caves and erosion pits in the Al-matrix. Figure 4c shows many river-shaped molten marks in the Mg-matrix. It is undoubtedly caused by continuous high temperature and high pressure generated by shock loading and unloading. Therefore, the Mg-matrix melts and then rapidly condenses to form a phase transition zone. The flat shape in the enlarged area of Figure 4c indicates that the magnesium alloy particles are deformed under compression. In addition, many small magnesium particles can be found, indicating that the recovery and recrystallization occur due to the high-strain compression.

Al/Mg interface in sample I has been completely delaminated (Figure 5), which proves the effect of interface on shielding performance. Figure 5a shows a boundary formed by the substances falling. EDS analysis indicates that the content of Al and Mg is 57.91% and 42.09%, respectively, suggesting that the substance is $Al_3Mg_2$ [18]. Figure 5b shows many dissociation steps and penetration cracks on the Mg-side fracture. Since cracks are generated near the crater wall by shock loading, it will arrive at the interface and rapidly propagate in the radial direction, resulting in weak performance due to the delamination of the interface. EDS results show that Al and Mg are 45.74% and 54.26%, respectively, indicating that it is mainly $Al_{12}Mg_{17}$ phase [18]. Therefore, the formation of brittle intermetallic compounds is the key to the interface performance, and then affects the service life of the Ti/Al/Mg density-graded materials.

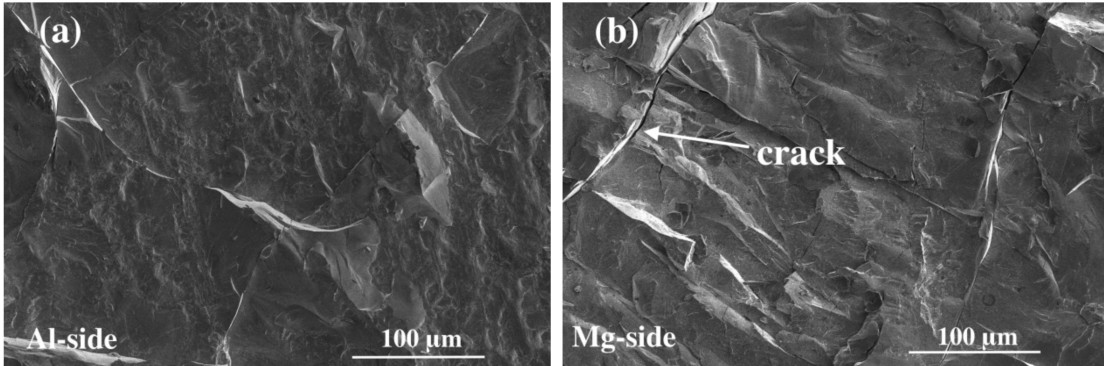

**Figure 5.** SEM micrographs of the interface for sample I. (**a**) Al-side; (**b**) Mg-side.

Figure 6 shows the micro-damage morphology of the crater in sample II. Due to the friction between aluminum alloy projectile and the steel barrel of gun, the EDS result of spheres on the crater wall shows that the main components are Al and Fe. Because the compression stress is greater than the compression strength of Ti-matrix, many cracks connected and the pieces fall off, as shown in Figure 6a. Therefore, the main damage mechanism is collapse damage created by the initial compression shock wave. Penetrating cracks in the Al-matrix are shown in Figure 6b. Because the titanium alloy layer fails to sufficiently dissipate the kinetic energy of the projectile, the aluminum alloy is damaged along cracks. Moreover, the softened magnesium alloy is exposed to high temperature and high pressure generated by shock wave loading and unloading, the tensile effect of shock wave causes the metal to splash, which reasonably explains the occurrence long jets in Figure 6c and the composition of the jets is the same with that of the Mg-matrix. In general, magnesium alloy is eroded and partly falls off until penetrated.

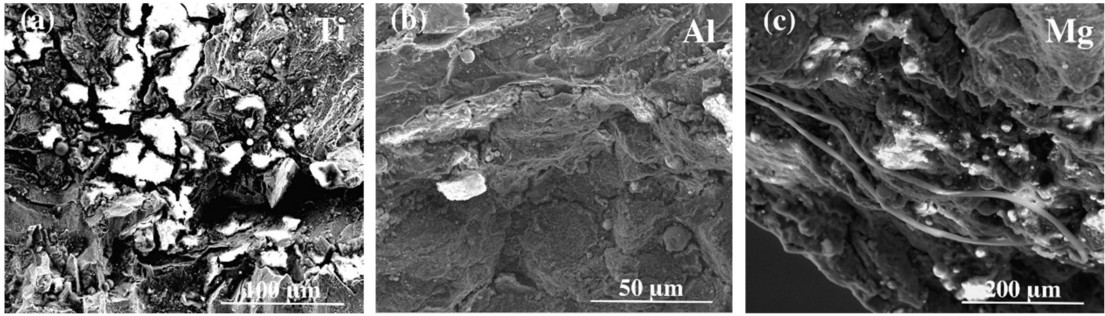

**Figure 6.** SEM micrographs of the crater for the sample II. (**a**) Ti-matrix; (**b**) Al-matrix; (**c**) Mg-matrix.

During the process of hypervelocity impact, since the heat generated near the crater wall is difficult to dissipate, the microstructure of the nearby substrate is significantly changed due to the increase of temperature. The micrographs of the matrix at 3 mm from the crater wall in sample II is shown in Figure 7. The structure shows some directional features. The compression shock wave is firstly generated and then causing tensile shock wave by reflecting on the surface of the bumper. Therefore, the layered feature in the Ti-matrix, the block in the Al-matrix and the stripe patterns in the Mg-matrix all point to the trailing edge of the crater. The Mg-matrix is severely compressed and the grains are severely deformed. No fusing phenomenon can be observed, indicating that the high temperature and pressure caused by the high strain rate compression of the shock wave are insufficient to melt the metal in this region.

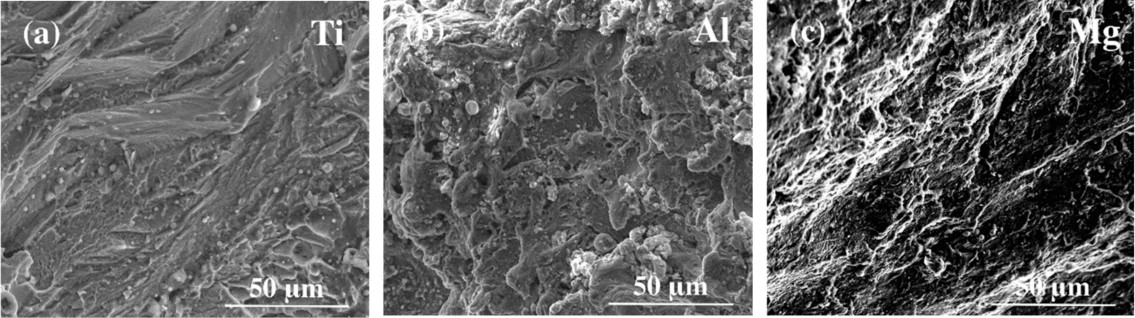

**Figure 7.** SEM micrographs of the matrix for the sample II. (**a**) Ti-matrix; (**b**) Al-matrix; (**c**) Mg-matrix.

### 4.3. Protection Mechanism

During the hypervelocity impact process, the shock wave oscillates many times in the system composed of the projectile and the protective structure, and it is accompanied by the transmission and transformation of the impact energy. Rankine and Hugoniot's mathematical method for shock waves is applied. Based on conservation of momentum, the reaction force F of the protective material to the projectile in the impact is obtained, and its expression is shown as follows [19]:

$$F = \rho_0 \times U_s \times U_p \times A \tag{1}$$

where $\rho_0$ is the density of the material under vacuum, kg/m$^3$; $U_s$ is the shock wave velocity, km/s; $U_p$ is the particle velocity, km/s; $A$ is the area of the impact point, m$^2$. It can be found that when the size and velocity of projectile are fixed, the main parameter determining the reaction force of the projectile is $\rho_0 \times U_s$, which is defined as the wave impedance ($R$). It is given by [19]:

$$R = \rho_0 C_0 + \rho_0 S U_p \tag{2}$$

where $C_0$ is the sound speed in the material under vacuum, km/s; $S$ is the coefficient in the Rankine-Hugoniot relationship. It seems that the wave impedance with small projectile velocity can be approximately equal to the product of density and sound speed. However, the Rankine-Hugoniot coefficient also has some influences on the wave impedance with the large projectile velocity. Key parameters of Ti/Al/Mg density-graded materials are listed in Table 1, which are taken from [20–24]. The wave impedance with the determined projectile velocity of each component can be calculated from the parameters in Table 1, and the calculated results show that the wave impedance is distributed from high to low in the impact direction.

**Table 1.** Key impact parameters of various components for Ti/Al/Mg density-graded material.

| Materials | Density (g/cm³) | $C_0$ (km/s) | $S$ | Melting Temperature (°C) | Vaporization Temperature (°C) |
|-----------|-----------------|--------------|------|--------------------------|-------------------------------|
| TC4 | 4.419 | 5.130 | 1.028 | 1800 | 3000 |
| 2A12 | 2.785 | 5.328 | 1.338 | 660 | 2057 |
| AZ31 | 1.78 | 4.516 | 1.256 | 651 | 1107 |

Figure 8 shows the typical morphology of debris clouds created by Ti/Al/Mg density-graded bumper and aluminum bumper at velocities close to 3.50 km/s. It confirmed that for the Al bumper, large fragments concentrated in the front edge of the debris cloud. However, sample I and sample II show a smaller largest central fragment of the debris cloud and a wider expanded area of projectile fragments. Outer-layer Ti of the Ti/Al/Mg density-graded bumper has higher wave impedance, which leads to higher peak of the impact pressure; thus, the projectile's velocity and penetration ability has been reduced. In addition, Ti/Al and Al/Mg interfaces can transmit shock waves and reflect unloading waves, which can be maintained for a certain time when equilibrium pressure is achieved, thereby the sustaining period and unloading period of the shock wave in the bumper is extended. Projectiles can be broken up into smaller parts, which greatly increases the entropy during the impact process, and more projectile kinetic energy is converted into the internal energy of the impact process. This explains the Ti/Al/Mg density-graded bumpers improve the protective ability by the dissipation of the kinetic energy via the debris cloud.

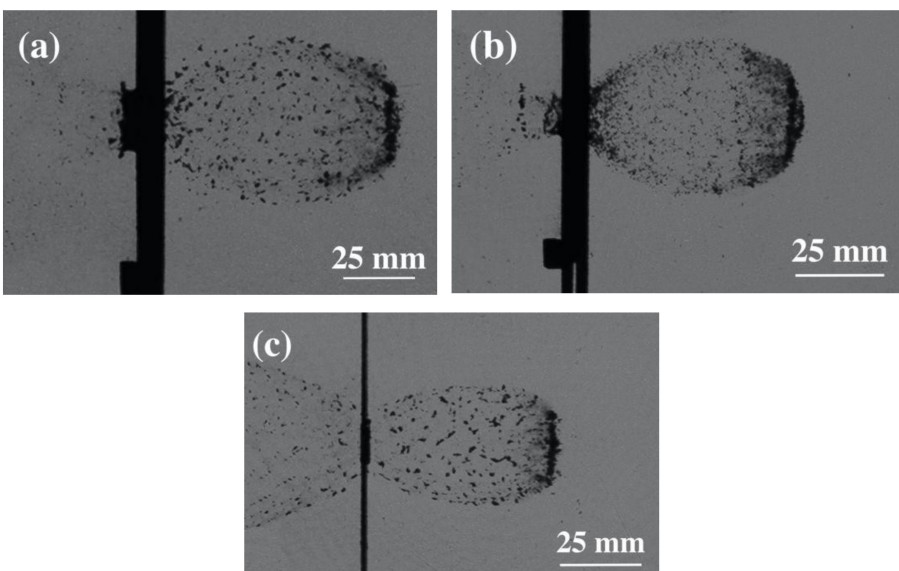

**Figure 8.** Late-time views of the debris clouds for different bumper. (**a**) bumper I; (**b**) bumper II; (**c**) aluminum alloy bumper.

In general, with the same impact parameters, the impact damage characteristics of the material are closely related to the matrix structure and interface state. Outside the crater of sample II, the Ti/Al and Al/Mg interfaces do not separate, and the interface state can effectively regulate the stress distribution

in the material. Therefore, the insufficient protection capacity of sample II is mainly attributed to the low strength of the matrix. The average tensile strengths of Ti, Al, and Mg are 769, 239 and 37 MPa, respectively, which are lower than those of sample I, 953, 359 and 187 MPa. It should be noted that low substrate strength suggests small wave impedance, and the low impedance medium cannot sufficiently break the projectile. As a result, the shock wave cannot be effectively attenuated. In this case, the protective performance of Ti/Al/Mg density-graded material is not excellent. Although sample I shows better shielding performance, its Al/Mg interface is completely delaminated, and the material overall failed by the local impact of the projectile, which limits the engineering application.

Overall, the subsequent research will focus on the following aspects. For sample I, better interface combination can absorb more debris kinetic energy, thereby improving the impact resistance of the material, it is necessary to ensure that areas outside the crater to play a protective role. Since the increased interface bonding strength improved the impact performance, the application field of Ti/Al/Mg density-graded material and its service life can be extended. Therefore, it is essential to further regulate the Al/Mg interface. For sample II, it should focus on matrix strengthening via adding trace elements, multi-stage solid solution and aging heat treatment, and optimizing the integrated sintering process combined with interface bonding and matrix densification. The high-strength matrix can increase the duration time of impact loading, and more kinetic energy of the projectile converted into the internal energy of the impact process. Meanwhile, the high-impedance material leads to the projectile sufficient break up, and reduces the incident depth, allowing a greater improvement in protective performance to be achieved.

Therefore, Ti/Al/Mg density-graded material with the maximized matrix strength and better bonding performance have highly likely excellent application prospects in aerospace, where light weight, high strength, and excellent protective performance can be achieved.

## 5. Conclusions

Hypervelocity impact experiments were performed to study the damage characteristics of Ti/Al/Mg density-graded materials that prepared by diffusion bonding and powder metallurgy, respectively. The general conclusions of the work presented in the current paper can be summarized as following:

(1) The impact crater of sample I was a reverse petal-shaped failure morphology showing typical characteristics of ductile materials. Sample II was collapsed directly by shock waves, and the crater was a regular circle. The Al/Mg interface of sample I was completely delaminated, while those of sample II were well bonded and exhibited significant mechanical synergetic response.

(2) A theoretical analysis was performed to explore the mechanism of protection and the propagation process of shock wave. Compared with aluminum bumper, the hypervelocity impact characteristics of Ti/Al/Mg density-graded shields showed a smaller largest central fragment of the debris cloud and a wider expanded area of projectile fragments. The results showed that the unique wave impedance gradient characteristics of Ti/Al/Mg density-graded shields could regulate the propagation path of the shock wave and increase the sustaining period, so the higher degrees of fragmentation achieved, which was beneficial to convert more projectile kinetic energy into the internal energy.

(3) It demonstrated the potential applications of Ti/Al/Mg density-graded materials in the orbital debris shields. The further research focused on the control of interface bonding strength and matrix strength should be performed.

**Author Contributions:** Conceptualization, L.L. and Y.P.; methodology, L.L. and Y.P.; validation, L.L., and W.Z.; formal analysis, Y.P.; investigation, L.L. and W.Z.; resources, W.L.; data curation, L.L.; writing—original draft preparation, L.L.; writing—review and editing, Y.P. and L.L.; visualization, L.L. and W.L.; supervision, Y.P.; project administration, L.L. and Y.P.; funding acquisition, L.L. and Y.P. All authors have read and agreed to the published version of the manuscript.

**Funding:** This research was financially supported by the Scientific Research Project from the Hunan Province Education Department (no. 18C0536, 17C0458), Special Funds for the Construction of Hunan Innovation Province

(grant no. 2019GK2052), and Special Fund for Changzhutan National Independent Innovation Demonstration Zone (grant no.2018XK2203). We would like to thank Editage(www.editage.cn) for English language editing.

**Conflicts of Interest:** The authors declare no conflict of interest.

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
