# Peer review of "Study on Hypervelocity Impact Characteristics of Ti/Al/Mg Density-Graded Materials"

_metals, doi:10.3390/met10050697_

Round 1
Reviewer 1 Report
This manuscript presents an interesting study on the hypervelocity impact on the Ti/Al/Mg graded materials. The microstructure and the characteristics of the bumpers and craters were provided and discussed. However, the introduction does not give enough background related to this study. The protection mechanism (section 4.3) does not seem convincing, i.e., simulation or calculation should be carried.
I also have some questions listed below.
- line 32-33, “At present,…blocked.” I do not think it is formal to write in this way. Maybe you can reword like; there have been a limited number of high-performance protective materials, …so it is necessary to develop a new alloy…
- line 33-35, “Studies have shown…of space debris.” Could you explain what is “the gradient materials,” and why they can protect spacecraft”? Relevant references should be discussed here.
- Line 35-37, “In previous studies…based on mechanical properties”. This is a very general statement, and relevant studies should be described.
- line 46, for “Ti-TC4, Al-2A12, Mg-AZ31”, could you provide their compositions?
- line 47, what are the particle size of the powders?
- line 61, what is “multi-stage grinding and polishing”? More details are needed.
- line 63-64, what is the size of the tensile specimen?
- line 66, what kind of camera did you refer to? i.e., the resolution or the optical microscopy?
- For all the SEM images, you should mention whether they are secondary electron image or backscatter electron image. The corresponding detector should be introduced in Section 2.
- line 145, 149, for those EDS results, did you use the point, line, or mapping measurement? “EDS analysis indicate that the content of Al and Mg is 57.91%...the substance is Al3Mg2”. Did you confirm this phase with XRD?
- Line 149-150, did you confirm the Al12Mg17 with XRD? I would recommend doing so.
- Line 184, why did you apply Rankine and Hugoniot’s mathematical method?
- Regards to the wave propagation process, could you provide some numbers based on the model you applied, i.e., the shock wave propagation time?
- For the conclusion part, I would suggest rewrite this part based on the comments above.
Author Response
Response to Editor and Reviewer
Dear professors:
We would like to thank the editor again for giving us the chance to resubmit the paper, and also thank the reviewers for giving us constructive suggestions which would help us deeply improve the quality of the paper. Here we submit a new version of our manuscript, which has been modified according to the editor’s and reviewers’ suggestions.
The following is the point-to-point response to the editor’s and reviewers’ comments.
General comments:
Reviewer #1:
This manuscript presents an interesting study on the hypervelocity impact on the Ti/Al/Mg graded materials. The microstructure and the characteristics of the bumpers and craters were provided and discussed. However, the introduction does not give enough background related to this study. The protection mechanism (section 4.3) does not seem convincing, i.e., simulation or calculation should be carried.
I also have some questions listed below.
1. line 32-33, “At present,…blocked.” I do not think it is formal to write in this way. Maybe you can reword like; there have been a limited number of high-performance protective materials, …so it is necessary to develop a new alloy…
2. line 33-35, “Studies have shown…of space debris.” Could you explain what is “the gradient materials,” and why they can protect spacecraft”? Relevant references should be discussed here.
3. Line 35-37, “In previous studies…based on mechanical properties”. This is a very general statement, and relevant studies should be described.
4. Line 46, for “Ti-TC4, Al-2A12, Mg-AZ31”, could you provide their compositions?
5. line 47, what are the particle size of the powders?
6. line 61, what is “multi-stage grinding and polishing”? More details are needed.
7. line 63-64, what is the size of the tensile specimen?
8. line 66, what kind of camera did you refer to? i.e., the resolution or the optical microscopy?
9. For all the SEM images, you should mention whether they are secondary electron image or with backscatter electron image. The corresponding detector should be introduced in Section 2.
10. line 145, 149, for those EDS results, did you use the point, line, or mapping measurement? “EDS analysis indicate that the content of Al and Mg is 57.91%...the substance is Al3Mg2”. Did you confirm this phase with XRD?
11. Line 149-150, did you confirm the Al12Mg17 with XRD? I would recommend doing so.
12. Line 184, why did you apply Rankine and Hugoniot’s mathematical method?
13. Regards to the wave propagation process, could you provide some numbers based on the model you applied, i.e., the shock wave propagation time?
14. For the conclusion part, I would suggest rewrite this part based on the comments above.
Answer:
Thank you again for the comments on this paper.
Reviewer #1:
1. line 33-35, we have changed the expression way.
At present, there have been a limited number of high-performance protective materials, such as Kevlar cloth, Nextel cloth, Beta cloth, so it is necessary to develop a new material.
2.line 35-39, we briefly explained what is the gradient materials and the protect mechanism.
Over the recent decades, the homogeneous single-layer sheet has been replaced by multi-layer density-graded material sheet, such as Ti/Al/nylon[10] and Al/Mg[11] . The density gradient variation in the thickness direction to achieve high levels of impedance mismatch, which has been reported to improve penetration resistance through by multiple interface reflections and transmissions[12,13].
3.Line 41-42, relevant studies based on mechanical have been described.
it mainly based on the mechanical point, which concerns impact vaporization, jetting process, stress-wave propagation and thermodynamic state of impact fragment [14,15].
4. Line 50,
“Ti-TC4, Al-2A12, Mg-AZ31” are traditional alloy grades, and their compositions are based on national standards. So there is no need to provide their compositions.
5. Line 55,
with particle size of the powders are 45μm, 58μm and 48μm respectively.
6. Line70, we have explained “multi-stage grinding and polishing”.
multi-stage grinding by 200, 600, 1000, 1500 and 2000 grit SiC paper, then polished by 3μm Al2O3.
7. Line 74,
Tensile properties of the specimens with 25mm length were determined based on GBT228-2002.
8. Line 76,
Canon EOS 5D4 digital camera to record the macro damage.
9. The corresponding detector was introduced in Section 2.
Line 71, interfaces are backscatter electron image.
Line 79, samples near the impact region of the projectile are secondary electron images.
10. Line 158, our EDS results use the mapping measurement
11. Line 159-163, We can confirm this phase with XRD which are taken from [18].
12. Line 199,
When there is an internal discontinuity (such as shock wave) in the fluid flow, the differential equation loses its meaning. It is necessary to use the physically existing relationship between the mechanical quantities on both sides of the discontinuity, namely the Rankine-Hugoniot condition as a special boundary condition,which was called Jumping conditions. It is necessary for the internal boundary conditions of the flow and to ensure the uniqueness of the solution. The Rankine-Hugonio equation describes the relationship between the state of the shock wave in a one-dimensional flow in a fluid and the one-dimensional deformation in a solid.
13.Line 215-233, the protection mechanism (section 4.3) were further explanation. Typical morphology of debris clouds created by Ti/Al/Mg density-graded bumper and aluminum bumper at velocities close to 3.50 km/s were analyzed.
14. Line 262-280, the conclusion has been rewritten based on the comments above.
Best regards
Yours sincerely,
Yingbiao Peng
Reviewer 2 Report
English needs to be improved as it is difficult to read in the current version. Sentences have adjectives in the place of nouns and other significant problems.
Procedures and material processing needs to be explained in greater detail.
Reference 13 apparently has some information about processing, but is not used until later to justify the identification of several intermetallics.
Sample II shows the formation of an intermediate layer at the interface, which apparently leads to microcracking, but is not identified or even acknowledged in the paper.
Author Response
Response to Editor and Reviewer
Dear professors:
We would like to thank the editor again for giving us the chance to resubmit the paper, and also thank the reviewers for giving us constructive suggestions which would help us deeply improve the quality of the paper. Here we submit a new version of our manuscript, which has been modified according to the editor’s and reviewers’ suggestions.
The following is the point-to-point response to the editor’s and reviewers’ comments.
General comments:
Reviewer #2:
- English needs to be improved as it is difficult to read in the current version. Sentences have adjectives in the place of nouns and other significant problems.
- Procedures and material processing needs to be explained in greater detail.
- Reference 13 apparently has some information about processing, but is not used until later to justify the identification of several intermetallics.
- Sample II shows the formation of an intermediate layer at the interface, which apparently leads to microcracking, but is not identified or even acknowledged in the paper.
Answer:
Thank you again for your comments on this paper.
- The Editage(www.editage.cn) has helped us for Englishlanguage editing. Some obvious mistakes were corrected in the revised manuscript.
- Line 49-60, procedures and material processing have been explained in greater detail.The Ti/Al/Mg density-graded material prepared via diffusion bonding in this study was recorded as sample I, and Ti-TC4 bar , Al-2A12 plate and Mg-AZ31 bar were used as raw materials. The Ti/Al joint was achieved at 650℃ for 2h under the pressure of 10MPa, while Al/Mg was bonded at 475℃ for 1.5h under the pressure of 6MPa. Sample II was prepared by powder metallurgy technology, which includes multi-step powder metallurgy pre-sintering and interfacial joining. Ti-6Al-4V mixed powder, Al-2A12 and Mg-AZ31 alloy powder with particle size of the powders were 45μm, 58μm and 48μm respectively. Ti-6Al-4V was presintered under hot-pressing condition under 5MPa at 1250℃ for 3 h,Al-2A12 was presintered at 550℃ for 2 h, while Mg-AZ31 was presintered at 525℃ for 4 h.
- Reference 13 is only refers to the Al/Mg joints, the detailed process parameters of Ti/Al/Mg density-graded material have been revised in Line 49-57.
- Our previous studies have shown that the generated phases in the Al/Mg interface are Al3Mg2and Al12Mg17 brittle intermetallic compounds (IMCs). A proper IMCs layer can maintain a good connection of the interface, while the IMCs layer is too thick, the performance of the interface decreases sharply. Line 103-105, Al/Mg interface of sample I completely delaminated by the lateral compression waves, while the sample II is well bonded, just a slight delamination exists near the crater. Due to interface separation, Sample I has been analyzed in Fig. Line 235-239, Outside the crater of sample II, the Ti/Al and Al/Mg interfaces do not separate, and the interface state can effectively regulate the stress distribution in the material. Therefore, the insufficient protection capacity of sample II is mainly attributed to the low strength of the matrix.
Best regards
Yours sincerely,
Yingbiao Peng
Reviewer 3 Report
This paper describes the hypervelocity impact characteristics of Ti / Al / Mg materials well. However, there is a lack of study on previous researches, and it is expected that a better paper will be made if this part and the points indicated are reinforced.
1. The introduction part needs to be reinforced with regard to the mechanical effects of diffusion bonding and powder metallurgy. And I recommend you change the introduction be more logical.
2. Please indicate strain rate for tensile test in 2.3 part. Since the difference in mechanical behavior will occur depending on the strain rate, it seems to be important to determine the range of the strain rate.
3. Figures 1, 2, 4, 5, 6, and 7 will be helpful for understanding the figure by adding a detailed description for each figure.
Author Response
Response to Editor and Reviewer Dear professors: We would like to thank the editor again for giving us the chance to resubmit the paper, and also thank the reviewers for giving us constructive suggestions which would help us deeply improve the quality of the paper. Here we submit a new version of our manuscript, which has been modified according to the editor’s and reviewers’ suggestions. The following is the point-to-point response to the editor’s and reviewers’ comments. General comments: Reviewer # 3: This paper describes the hypervelocity impact characteristics of Ti / Al / Mg materials well. However, there is a lack of study on previous researches, and it is expected that a better paper will be made if this part and the points indicated are reinforced. 1. The introduction part needs to be reinforced with regard to the mechanical effects of diffusion bonding and powder metallurgy. And I recommend you change the introduction be more logical. 2.Please indicate strain rate for tensile test in 2.3 part. Since the difference in mechanical behavior will occur depending on the strain rate, it seems to be important to determine the range of the strain rate. 3.Figures 1, 2, 4, 5, 6, and 7 will be helpful for understanding the figure by adding a detailed description for each figure. Answer: Thank you again for your comments on this paper. 1. Line 234-245, mechanical effects of diffusion bonding and powder metallurgy has been discussed. So these didn't involve in the introduction part. In general, with the same impact parameters, the impact damage characteristics of the material are closely related to the matrix structure and interface state. Outside the crater of sample II, the Ti/Al and Al/Mg interfaces do not separate, and the interface state can effectively regulate the stress distribution in the material. Therefore, the insufficient protection capacity of sample II is mainly attributed to the low strength of the matrix. The average tensile strengths of Ti, Al, and Mg are 769MPa, 239MPa and 37MPa, respectively, which are lower than those of sample I, 953MPa, 359MPa and 187MPa. It should be noted that low substrate strength suggests small wave impedance, and the low impedance medium cannot sufficiently break the projectile. As a result, the shock wave cannot be effectively attenuated. In this case, the protective performance of Ti/Al/Mg density-graded material is not excellent. Although sample I shows better shielding performance, its Al/Mg interface is completely delaminated, and the material overall failed by the local impact of the projectile, which limits the engineering application. Line 30-46, we have changed the expression way in the introduction. With the rapid development of aerospace technology, the debris environment has become worsening, and the service life of orbiting spacecraft has been greatly restricted [1]. For space debris of millimeter level and below, protective structures composed of high-tech materials can be set in several key parts [2]. At present, there have been a limited number of high-performance protective materials, such as Kevlar cloth, Nextel cloth, Beta cloth, so it is necessary to develop a new protective material [3-9]. Over the recent decades, the homogeneous single-layer sheet has been replaced by multi-layer density-graded material sheet, such as Ti/Al/nylon [10] and Al/Mg [11] . The density gradient variation in the thickness direction to achieve high levels of impedance mismatch, which has been reported to improve penetration resistance through by multiple interface reflections and transmissions [12,13]. In previous studies on the hypervelocity impact characteristics of materials, most of the studies concern traditional titanium, aluminum, iron and flexible materials, and it mainly based on the mechanical point, which concerns impact vaporization, jetting process, stress-wave propagation and thermodynamic state of impact fragment [14,15]. In this study, Ti/Al/Mg density-graded materials were prepared by diffusion bonding and powder metallurgy, respectively. Deformation and damage characteristics of these materials under hypervelocity impact were systematic analyzed. The impact damage mechanism of Ti/Al/Mg density-graded materials can be revealed by damage morphologies. 2. Line 74-75,Tensile properties of the specimens with 25mm length were determined based on GBT228-2002,with a loading rate of 1 mm/min. 3. Detailed description for each figure in the corresponding text: Line 49, Ti/Al/Mg density-graded material prepared via diffusion bonding in this study was recorded as sample I. Line 53, Sample II was prepared by powder metallurgy technology. Line 49-57, procedures and material processing have been explained in detail. So fig.1 just described as SEM micrographs at the interface of Ti/Al/Mg density-graded materials. Line 94-95, The macro-scale damage patterns of samples I and II are compared in Fig.2, where A and B denote the front surface and back surface, respectively. Impact testing parameters in Line 65-67. The detailed description for each figure was in Line 120, Line 156, Line 168, Line 184, respectively. Best regards Yours sincerely, Yingbiao PengReviewer 4 Report
There are several issues in the manuscript that should be addressed before further consideration for publication. 1. Did the authors consider other form of joining, such as additive manufacturing? - Tey et al. (2020), Additive manufacturing of multiple materials by selective laser melting: Ti-alloy to stainless steel via a Cu-alloy interlayer, Additive Manufacturing 31, 100970 - Chen et al. (2020), Selective laser melting 316L/CuSn10 multi-materials: Processing optimization, interfacial characterization and mechanical property, Journal of Materials Processing Technology 283, 116701 2. In materials preparation, clarify if the compositions used for both methods are the same? What are the exact compositions? How is the methods parameters chosen? 3. For the intermetallics formed, any further analysis conducted? For example, using XRD or TEM to confirm their phase? 4. Suggest to do in-depth analysis of the fracture surfaces to identify dislocations of the secondary phases etc.Author Response
Response to Editor and Reviewer Dear professors: We would like to thank the editor again for giving us the chance to resubmit the paper, and also thank the reviewers for giving us constructive suggestions which would help us deeply improve the quality of the paper. Here we submit a new version of our manuscript, which has been modified according to the editor’s and reviewers’ suggestions. The following is the point-to-point response to the editor’s and reviewers’ comments. General comments: Reviewer # 3: 1.Did the authors consider other form of joining, such as additive manufacturing? - Tey et al. (2020), Additive manufacturing of multiple materials by selective laser melting: Ti-alloy to stainless steel via a Cu-alloy interlayer, Additive Manufacturing 31, 100970 - Chen et al. (2020), Selective laser melting 316L/CuSn10 multi-materials: Processing optimization, interfacial characterization and mechanical property, Journal of Materials Processing Technology 283, 116701. 2.In materials preparation, clarify if the compositions used for both methods are the same? What are the exact compositions? How is the methods parameters chosen? 3.For the intermetallics formed, any further analysis conducted? For example, using XRD or TEM to confirm their phase? 4.Suggest to do in-depth analysis of the fracture surfaces to identify dislocations of the secondary phases etc. Answer: Thank you again for your comments on this paper. 1.Our earlier studies have proved that intermetallic compounds are the key to interfacial performance, and it is very sensitive to temperature changes. Additive manufacturing is an excellent method for interface connection, while intermetallic compounds near the interface grow rapidly at the high process temperature, resulting in the sharp decreased of interface performance. In addition, when solder is added, the interface density will change abruptly, changing the original density gradient, thereby affecting the distribution of wave impedance, and ultimately affecting the shielding performance. 2.Line 49-60, procedures and material processing have been explained in greater detail. The Ti/Al/Mg density-graded material prepared via diffusion bonding in this study was recorded as sample I, and Ti-TC4 bar, Al-2A12 plate and Mg-AZ31 bar were used as raw materials. The Ti/Al joint was achieved at 650℃ for 2h under the pressure of 10MPa, while Al/Mg was bonded at 475℃ for 1.5h under the pressure of 6MPa. Sample II was prepared by powder metallurgy technology, which includes multi-step powder metallurgy pre-sintering and interfacial joining. Ti-6Al-4V mixed powder, Al-2A12 and Mg-AZ31 alloy powder with particle size of the powders were 45μm, 58μm and 48μm respectively. Ti-6Al-4V was presintered under hot-pressing condition under 5MPa at 1250℃ for 3 h,Al-2A12 was presintered at 550℃ for 2 h, while Mg-AZ31 was presintered at 525℃ for 4 h. 3.Line 159-163, We can confirm this phase with XRD which are taken from [17,18]. 4.Line 159, XRD and TEM of the secondary phases were deeply analyzed in [18], so the duplicate data wasn’t further explained. Best regards Yours sincerely, Yingbiao PengRound 2
Reviewer 1 Report
The authors have answered the questions and made changes accordingly.
I would like to accept the current version of the manuscript.
Reviewer 2 Report
English program resulted in much improved readability. Still some more minor English problems.
Reviewer 4 Report
There are some minor errors in the manuscript that should be edited.
- Line 12 - types instead of typs